# Mobile Sources Mixing Model Implementation for a Better Quantification of Hydrochemical Origins in Allogenic Karst Outlets: Application on the Ouysse Karst System



**David Viennet [1],\*, Guillaume Lorette [1,2], David Labat [3], Matthieu Fournier [4], Mathieu Sebilo [5], Olivier Araspin [6] and Pierre Crançon [6]**

1 Causses du Quercy UNESCO Global Geopark, Labastide-Murat, 46240 Cœur de Causse, France
2 Syndicat Mixte des Eaux de la Dordogne, 24430 Marsac sur l'Isle, France
3 Géosciences Environnement, University of Toulouse Toulouse, CNRS UPS IRD CNES, 31400 Toulouse, France
4 Continental and Coastal Morphodynamics Laboratory Normandie University, UNIROUEN, UNICAEN, CNRS M2C, 76000 Rouen, France
5 Institute of Ecology and Environmental Sciences–Paris, Sorbonne Université, CNRS, INRAE, IRD, UPD, UPEC, IEES, 75005 Paris, France
6 CEA, DAM, DIF, F-91297 Arpajon, France
\* Correspondence: dviennet@parc-causses-du-quercy.org

**Abstract:** On the edge of sedimentary basins, karst aquifers can be fed by several water sources from both autogenic and allogenic recharge. In some cases, assessing water origins can be hard and cause some difficulties for water resource management. The main goal of this study is to show the implementation of the mobile sources mixing model approach. More precisely, this research develops how a monitoring method using a multi-proxy approach can be used to quantify waters sources contributions from several origins at the outlets of a karst system. The study site is the Ouysse karst system, located in western France. The site offers the opportunity to understand the mixing processes between allogenic and autogenic water recharges. The karst system covers a 650 km$^2$ watershed, and is fed by three different chemical facies: (i) Autogenic water from the direct infiltration on the karstified limestones with high HCO$_3^-$ values (median: 436 mg.L$^{-1}$); (ii) Water coming from sinking rivers fed by spring coming from igneous rocks with low mineralization but relatively higher K$^+$ values (median: 4.2 mg.L$^{-1}$); (iii) Highly mineralized water coming from deep evaporitic layers and feeding another sinking river with very high sulfate concentrations (median: 400 mg.L$^{-1}$). Sliding window cross-correlation analyses and hydrochemical analyses during a flood event are performed to implement a mobile source mixing model approach. This approach shows significant differences with a simple fixed source mixing model and appears more reliable but requires more time and money to carry out. The results and conclusion of this study will be used for forecasting and managing operational actions for resource management.

**Keywords:** karst aquifer; hydrochemistry; hydrodynamic; hydrogeology; source mixing model; cross correlation; water resource management

## 1. Introduction

Among groundwater bodies, karst aquifers are widely used as a drinking water resource. Carbonate rocks are widely distributed throughout the world, particularly in Europe [1]. Hence, 20% to 25% of the world's population relies largely, or entirely, on karst aquifers [2,3].

The formation of a karst system driven by carbonate dissolution leads to considerable heterogeneity of both surface and the underground environment with an organization of the voids determined by the underground flows [4]. Because of these unique hydrogeological features, rapid transfers can occur between sinking rivers and groundwater resources.

Thus, karst aquifers are vulnerable to anthropogenic activities and require specific management [5]. Water resources management in karst areas can only be effective with a good knowledge of the structure of karsts and hydraulic transfers. A multidisciplinary approach combining hydrodynamic and natural hydrochemical responses quantification is needed to reach a global characterization of the dynamics of karst aquifers [6–17].

This quantification is made by the geochemical decomposition of chemiographs. Usually, the decomposition is made with fixed characteristic concentrations attributed to each source/end member, to estimate their respective contributions in the mix at the outlets of the system [17–23]. These concentrations are generally fixed because their temporal variations are small. This is the case for (i) poorly karstified aquifers, where percolating water has time to balance itself with the epikarst and unsaturated zone before reaching the saturated zone; (ii) karst aquifers fed by a deep confined aquifer with constant chemistry [24]. In the case of highly connected karst aquifers like the Ouysse system (Occitanie, France), water transfers are too fast to be balanced before reaching the saturated zone. Thus, the water chemistry shows wide variations for each origin in a mix due to temporal variations in the proportions in the mix [17].

To better take these variations into account, the main goal of this paper is to implement a "mobile" sources mixing model approach.

The Ouysse karst aquifer offers the possibility of understanding the mixing processes by monitoring both hydrodynamic responses and natural hydrochemical tracers. These processes are studied with the mobile sources mixing model approach. Hence, it is a chance to "open up the black box" of the sink-to-spring continuum, with a direct look at the underground river linking the two.

To achieve this goal, a multi-proxy approach was carried out in this work. It is decomposed in the following steps:identifying groundwater flow paths, water origins, and their respective contributions to outlets in a complex karst aquifer with both allogenic and autogenic recharge. This article presents the set-up of an environmental observatory dedicated to karst systems and shows its use in the assessment of their dynamics and vulnerability to external forcing.

The first step is to evaluate the time lag between each sinking stream and outlet. The second one is to assess the contribution of each water origin to the outlets of the system using the mobile sources model. Then, the third one is to compare this innovative approach with a fixed source model mixing.

## 2. Context of the Study Area

The Ouysse karstic system drains a mix of allogenic and autogenic waters, located at the north-east limit of the Aquitaine sedimentary basin (66,000 km$^2$). It is located in the northern part of the Regional Natural Park and UNESCO Global Geopark of the Causses du Quercy and supplies drinking water to about 80,000 inhabitants. This water extraction spreads over 3 pumping sites: (i) the Courtilles borehole, capturing the main underground river of the karstic system; (ii) the main outlet called Cabouy spring; and (iii) the Fontbelle outlet, a minor spring downstream from Cabouy.

### 2.1. Geology and Hydrogeology

The Ouysse karst system geological setting can be described as a tabular sequence from Lower to Middle Jurassic carbonate formations, with a thickness of 300–350 m, resting upon an igneous Paleozoic substratum (Figure 1).

This bedrock is mainly composed of gneiss, schist, and micaschist. Above, 50 m of Triassic detrital sediments were deposited. At the top of these formations (bottom of the Lower Jurassic), an evaporitic gypsum layer can be found. Then marls and carbonates are found and compose the Lower Jurassic layers. The outcropping area of these marls is called the Limargues. Then, the Middle Jurassic carbonate sequence starts (Bajocian, Bathonian, Callovian, and Oxfordian), consisting of dolomitic limestones (Figure 1). The outcropping area of these karstified limestones is called the Causse. The whole sedimentary sequence is

slightly tilted toward the southwest, so the youngest outcropping formations are located in the west of the hydrogeological catchment, to the center of the sedimentary basin, and the older rocks are found at the east.

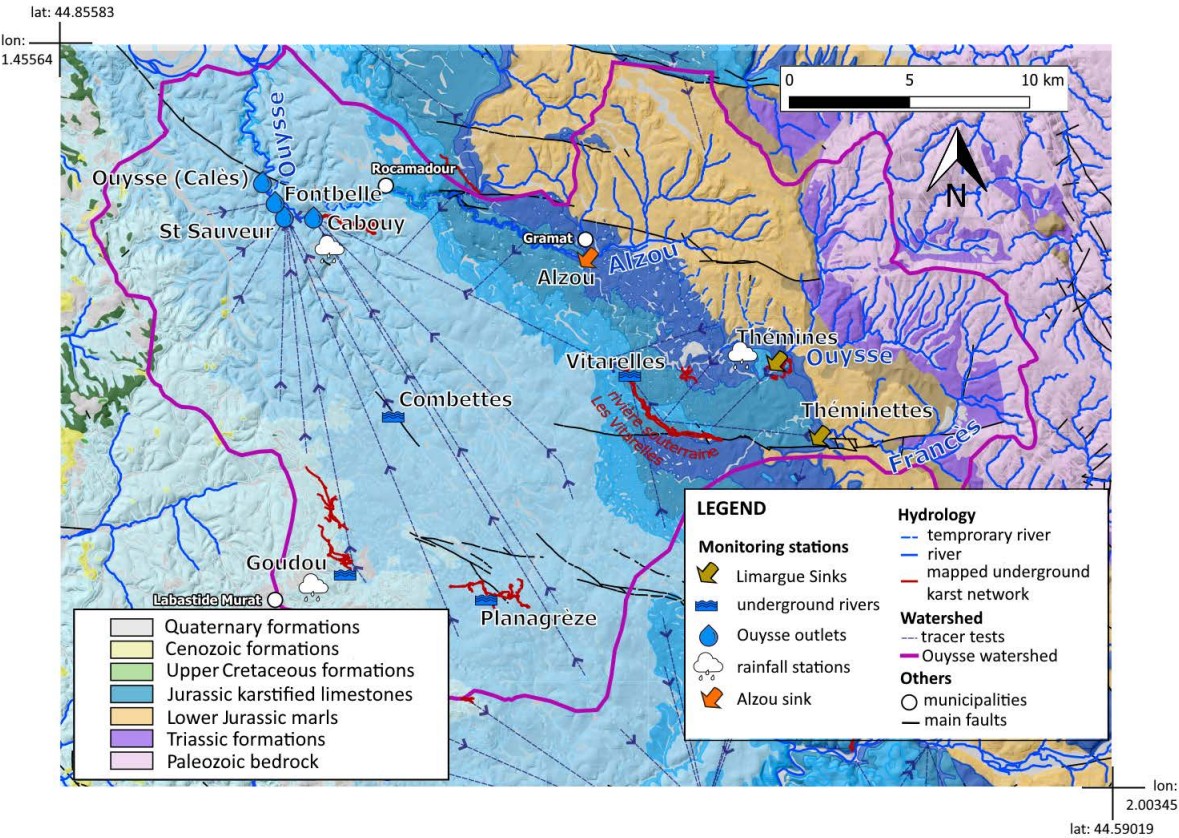

**Figure 1.** Geological map of the study area and location of the monitoring stations.

The geomorphology of the site is a typical karst landscape with numerous sinkholes and swallow holes at the surface of the carbonate outcrops.

The Ouysse karst system has a watershed of 650 km$^2$, mixing autogenic and allogenic water. The allogenic part is an area of 154 km$^2$ corresponding to a runoff basin whose surface water is concentrated on several streams which then leads the water into the karst system in several sinks as the rivers reach the eastern boundary of the karstic carbonate formations.

The whole Jurassic carbonate sequence is considered one main aquifer (Bajocian, Bathonian, Callovian, and Oxfordian). Water flows out of the system through three main outlets: the Cabouy spring, the Saint-Sauveur spring, and the Fontbelle spring. The downstream Ouysse river emerges from these three springs. Several decades of tracer tests allowed to define the limits of the watershed (Figure 1) and study the water transfer between the sinking streams, the underground rivers, and the outlets, showing transfer times between 26 m h$^{-1}$ and 135 m h$^{-1}$ [25].

### 2.2. Hydrological and Hydrogeological Context

The watershed catches three perennial rivers: the Alzou, the Ouysse, and the Francès. All three flow through the Limargue area (where the marls are outcropping) and sinks as they reach the outcropping carbonates of the Causse area. The catchment areas of the Francès and Ouysse (southeast of the watershed) are mainly runoff from the Limargues (marl) and Segala (Paleozoic bedrock) zones. The Alzou River, located further north in the watershed, is partly fed by a spring that emerges through a fault in the middle of the Limargue marl layers. This water comes from the evaporite aquifer located at the base of the Lower Jurassic and is highly mineralized. (Figure 1). The three rivers show an important

variation of flow according to the rain events: the smallest is the Francès river flowrate varying between 1 L.s$^{-1}$ to 5000 L.s$^{-1}$, the Alzou river flowrate varies between 20 L.s$^{-1}$ and 8000 L.s$^{-1}$, and finally, the Ouysse flowrate varies from 20 L.s$^{-1}$ to 10,000 L.s$^{-1}$ [17]. Several tracer tests have identified the connections between the losses of these three rivers and the springs of Cabouy, Saint Sauveur, and Fontbelle [25].

The infiltration of water on a single point allowed by the sinking rivers has strongly accelerated the karstification process in the limestones of the Middle and Upper Jurassic. This contributed to the creation of numerous underground rivers. In the Ouysse karst system, the most important one (with wide drains from 20 to 70 m in diameter) is the underground river of Vitarelles, exploited for drinking water via the Courtilles Borehole. Tracer tests have revealed that the Francès river and the Ouysse river sinks feed this underground river before connecting with the outlets.

The hydrogeological dynamic of the Ouysse karst system has been described in previous work [17,25–28]. This system is an example of a mixed autogenic and allogenic karst system. The allogenic sinking rivers recharge can feed the system significantly, especially during low flow periods. During high flow, it is the autogenic karst area which brings the majority of the hydrochemical contributions [17].

Infiltrations into the sinks may lead to a degradation of water quality in underground rivers [17]. Thus, the precise quantification of the proportion of surface water is necessary to ensure good management of the water resource, in particular in the drinking water catchments (Courtilles, Cabouy, Fontbelle). In view of the great variability of water from different origins, it was necessary to adapt the decomposition of the hydrograph with mobile end members.

## 3. Materials and Methods

The data used in this study comes from several years of monitoring the Ouysse karst system inlets, outlets, and underground networks.

### 3.1. Continuous Data Monitoring and Discrete Sampling

Since July 2017, hydrodynamic data (water level, temperature, electric conductivity, and rainfall) are monitored in sinking streams, underground rivers, and main outlets of the Ouysse karst system.

Rainfall is measured on three weather stations located on strategic places over the hydrogeological catchment. Only accumulated rainfall over an hour for the central station is presented in this article.

For sinking-stream, water-level (WL), electrical conductivity (EC) and temperature (T) are recorded since March 2018, December 2018 and September 2019 for Alzou, Thémines, and Theminettes, respectively. For underground rivers, the same parameters are recorded since October 2017. For outlets, these parameters are recorded since July 2017 at Cabouy spring, December 2018 at St-Sauveur spring, and Fontbelle spring. They are recorded hourly using CTD divers with an accuracy of $\pm 0.5$ mm (WL), $\pm 1$ μS.cm$^{-1}$ (EC), and $\pm 0.1$ °C (T).

In June 2020, five Aquatroll600 were installed at Thémines and Alzou sinking streams and at Cabouy, St-Sauveur, and Fontbelle springs. In addition to the previous CTD parameters, it allows the measurement of turbidity, dissolved oxygen (DO), and chlorophyll-A at hourly steps.

Water sampling on the catchment area of the Ouysse karst system has been carried out at least once a day (twice during flood peak) from 30 September 2020 to 27 October 2020. Full detailed results of this monitoring are shown in Appendix B. On-field filtration (0.45 μm) of the water samples was carried out, and acidification with nitric acid (HNO$_{3-}$ 70% pure) was applied for cation measurement. Alkalinity (HCO$_3$$^-$) was directly measured on-site by titration.

The measurement of major ions concentrations (Ca$^{2+}$, Mg$^{2+}$, Na$^+$, K$^+$, HCO$_3$$^-$, Cl$^-$, SO$_4$$^{2-}$, NO$_3$$^-$) was carried out by ICP-MS.

*3.2. Methods*

To reach the goals of this article, two methods will be applied to this data: (i) sliding window cross-correlation analyses, a method developed by Delbart et al. [29], and (ii) mobile source model mixing, an innovative method for improving the model mixing proposed by Lorette et al. [17].

### 3.2.1. Correlation Analyses on Hydrological Time Series

Cross-correlation analyses (CCA) are widely used in signal processing. These analyses can be used to characterize the temporal structure of a hydrological signal [30–34]. Some authors have used CCA to determine the relationships between several variables, such as rainfall, flow rates, electrical conductivity, turbidity, or water temperature [32,35–38].

When rainfall and discharge are considered respectively as the input and output signals, the correlogram, memory effect, and time lag (pressure wave transfer) can be used to describe the degree of karstification of the aquifer and its response to rainfall events [39–41]. When the conductivity of a sink and spring are respectively considered as the input and output signals, the time lag can be used to determine the mass transfer time, and thus be used for source mixing calculation.

The sliding window cross-correlation analysis is an innovative approach developed by Delbart et al. [29,42]. This approach allows us to study the temporal evolution of the correlation between input and output. The karst is thus no longer considered a stationary entity. Indeed, karst aquifers generally present a complex response due to their spatial heterogeneity. The purpose of this method is to place this temporal variability in the local seasonal hydrological cycle in order to determine if it has an influence on the response time.

A measurement window of a fixed time period, adequate to the dataset considered, is determined by the user (Figure 2). In this case, the window was set to four months, selected to take into account the seasonal cycle observed at each station. A cross-correlation analysis is performed on this window, which then shifts by two months to analyze the following data sequence. This overlapping allows the data from each window to be partially reanalyzed and interpreted with the next window. Each data will thus be analyzed by two different windows (except at the beginning and end of the chronicle).

### 3.2.2. Source Mixing Calculation

The water volume in a mix ($V_f$) of two end members ($V_1$ and $V_2$) in a stationary mass transfer is the sum of all end members ($V_f = V_1 + V_2$). If two non-covariant variables of concentration $A$ and $B$ are present in each end member, the mixing equations are as follows (Equation (1)):

$$\begin{cases} \alpha.A_1 + \beta.A_2 = A_f \\ \alpha.B_1 + \beta.B_2 = B_f \end{cases} \text{ with } \alpha = \frac{V_1}{V_f} \text{ and } \beta = \frac{V_2}{V_f} \tag{1}$$

It is possible to adapt these mass balance equations for more than two origins: with a mixture $f$ with $n$ poles and $K$ non-covariant variables: (Equation (2)):

$$\begin{pmatrix} \alpha \\ \vdots \\ \eta \end{pmatrix} = \begin{pmatrix} A_1 & \cdots & A_n \\ \vdots & \ddots & \vdots \\ K_1 & \cdots & K_n \end{pmatrix} \begin{pmatrix} A_f \\ \vdots \\ K_f \end{pmatrix} \tag{2}$$

$A_1, \ldots, K_n$ are the concentrations of the variables ($A$ to $K$) for every end member (1 to $n$). $A_f, \ldots, K_f$ are the values for each variable in the mix. Solving this matrix equation leads to know $\alpha, \ldots, \eta$, the contributions of every end member to the chemical balance of $f$.

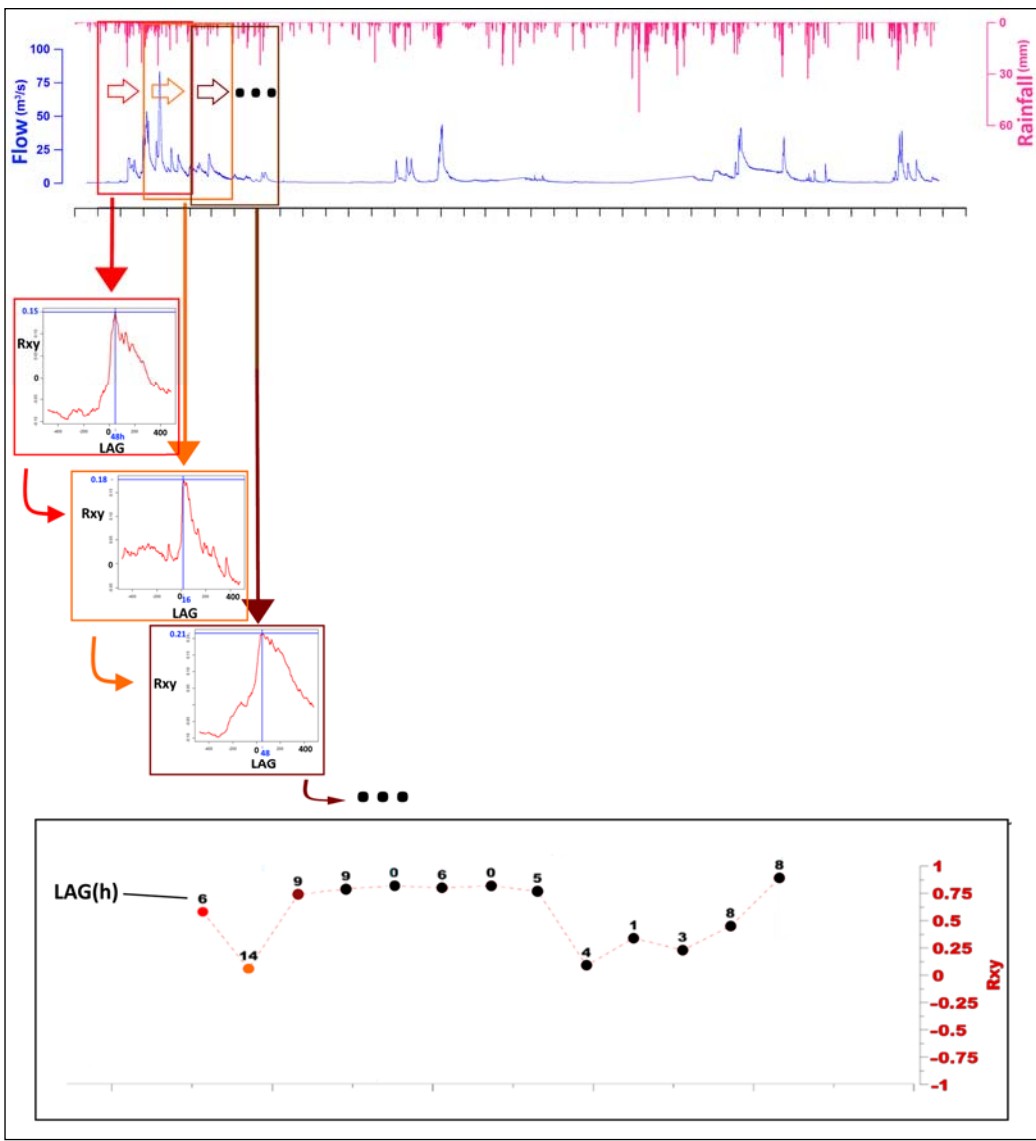

**Figure 2.** Illustration of a sliding window CCA between rainfall and outlet flow, adapted from Delbart.

### 3.3. The Mobile Sources Mixing Model Approach

Assessment of the contribution of each water origin to the outlets of the Ouysse system can be done using fixed concentrations of sources [17–22]. According to the hydrogeological dynamic in karst systems this simple method needs to be adapted because of the high temporal variability of water composition for each source [17]. In this work, a mobile source mixing model approach is implemented using a simple acquisition of hydrodynamical and hydrochemical parameters.

The sliding windows cross-correlation analysis is applied to geochemical data to assess the average mass transfer time lag between each source and outlet. Then, the high-resolution monitoring during a flood is necessary to evaluate water quality variability of each origin. Most pertinent variables are selected according to the results of Lorette et al. [17] for the Ouysse karst system. Finally, a source mixing calculation is performed to quantify for each mixed sample the contribution of each endmember with its variable values (Equation (2)), with an offset corresponding to the duration of the water mass transfer time estimated with the cross-correlation methods (Figure 3).

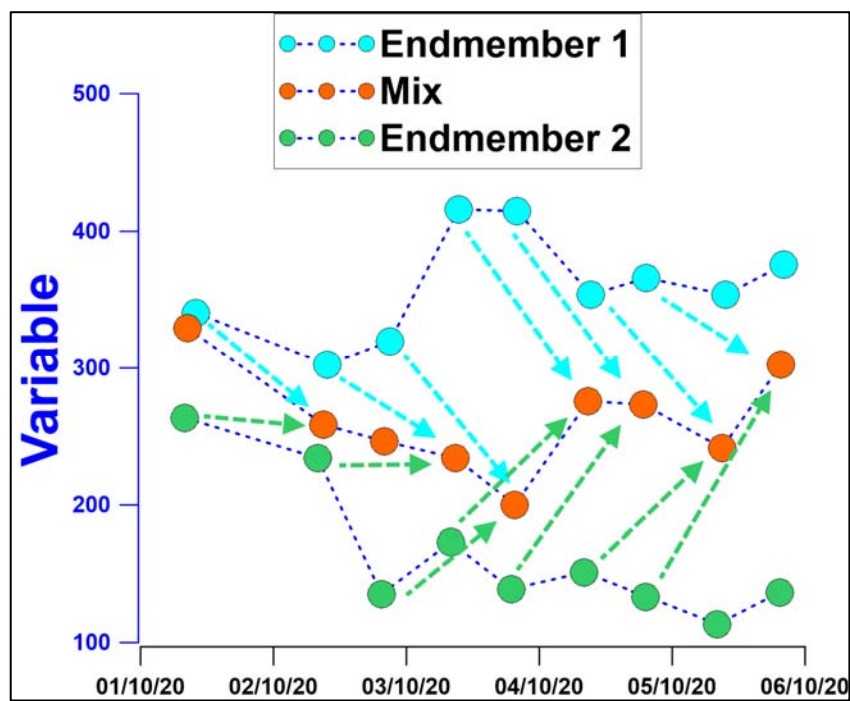

**Figure 3.** Illustration of how the mobile source mixing model works: the arrows represent an example of time lag of 1.5 day at which the water mass might transfer from all endmembers to the mix.

## 4. Results and Discussion

The previously illustrated methods were applied to the data obtained by monitoring the Ouysse karst system.

### 4.1. Sliding Windows Cross-Correlation Analyses on Long-Term Monitoring Results

Table 1 shows the great variability of the different points of the watershed. The rivers at the Thémines and Théminettes sinks show a relatively low conductivity, typical of water coming from a crystalline rock substratum. On the other hand, the Alzou sink, fed by a confined evaporitic aquifer, has a much higher conductivity. All three sinks have similar temperature variations. Themines has a higher mean flow rate than Theminettes and Alzou, with similar and lesser values. At the outlets, the conductivity and temperature are very similar. The Cabouy spring shows a greater variability, while Saint Sauveur and Fontbelle are much more stable. The downstream Ouysse river fed by the springs shows a mix of Cabouy, Saint-Sauveur, and Fontbelle behaviors. Its median flow rates are four times higher than the global flow of the sinks. The supplementary water volume is thus from an autogenic origin, infiltrated directly by the karst plateau.

#### 4.1.1. Crosscorrelation between Sinks and Springs Waterlevel

The cross-correlation between the different water levels measured at each station highlights the modalities of the pressure wave transfer in the karst system. The pressure wave corresponds to the arrival of water already present in the aquifer and pushed towards the exits of the system due to the arrival of fresh water by the "piston effect".

Figure 4 compares the water levels at the Fontbelle spring (in orange), and respectively at Alzou, Thémines, and Théminettes sinks (in gray). The rainfall is represented in blue. The water level of each sink is correlated to the water level of the springs and the Ouysse river at the Calès bridge, with respective delays. The time lag and the correlation value are calculated for the whole data set but also for each sliding window. Full graphical results for each station are shown in Appendix A.

**Table 1.** Summary of statistics on the flow/water level, temperature, and conductivity at each monitored station.

| | Flow (L/s) | | | Temperature (°C) | | | Conductivity (µS/cm) | | | Waterlevel (cm) | | |
|---|---|---|---|---|---|---|---|---|---|---|---|---|
| | Min | Median | Max | Min | Median | Max | Min | Median | Max | Min | Median | Max |
| Theminettes | <10 | 247 | 14,000 | 4.1 | 12.9 | 22.9 | 78 | 387 | 662 | / | / | / |
| Themines | <10 | 196 | 29,200 | 1.7 | 12.3 | 23.2 | 130 | 330 | 945 | / | / | / |
| Alzou | 20 | 160 | 10,800 | 4.5 | 13.5 | 23 | 262 | 1311 | 2145 | / | / | / |
| Cabouy | / | / | / | 10.5 | 13.3 | 16.5 | 218 | 563 | 847 | 30 cm | 72 cm | 196 cm |
| Saint Sauveur | / | / | / | 10.6 | 13.3 | 19 | 489 | 609 | 788 | 54 cm | 110 cm | 227 cm |
| Fontbelle | 0 | 137 | 1375 | 12.8 | 13.4 | 14 | 501 | 599 | 857 | / | / | / |
| Ouysse Cales | 454 | 2218 | 94,200 | 9.5 | 13.2 | 18.8 | 351 | 571 | 984 | / | / | / |

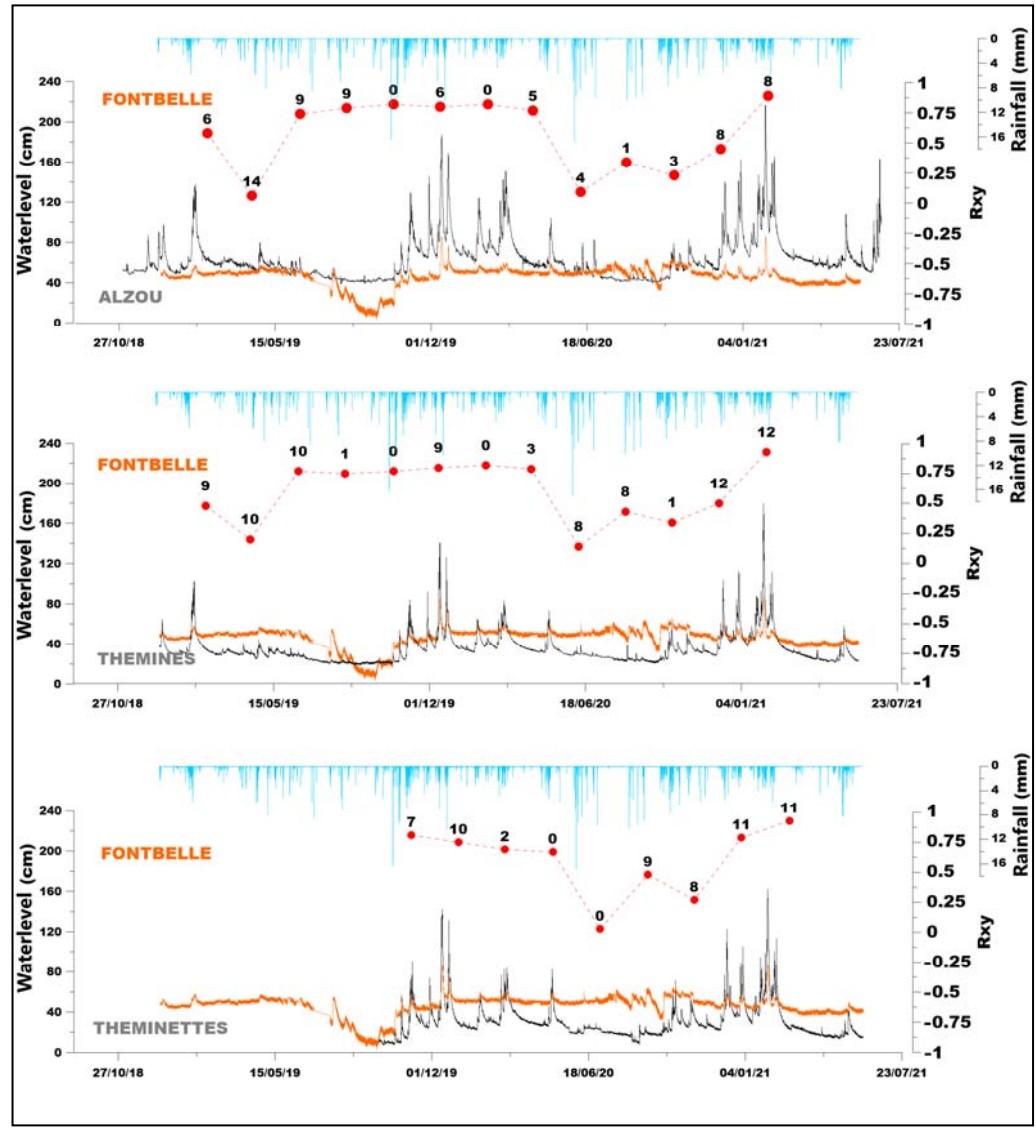

**Figure 4.** Sliding correlation analyses between the water levels for the Fontbelle station and the different stations at the sinks; the results of the sliding cross-correlation analyses are displayed in red dots (vertical axis: correlation, label: the time lag between the two sites in hours).

Statistic properties of lag and correlation values, determined from cross-correlation on each sliding window, are given in Table 2 and compared to the "global" lag value determined on the whole chronicle for each station (these results were already presented in the previous work [17]), that is considered to remain constant over the observation period.

**Table 2.** Compared results of the global CCA and the sliding window CCA between sinks waterlevel and springs waterlevel.

| CABOUY | Global CCA | | Median and [Min–Max] of Sliding Windows CCA | |
|---|---|---|---|---|
| | Lagmax (h) | correlation (Rxy) | Lagmax (h) | correlation (Rxy) |
| **ALZOU** | 8 | 0.4 | 8 [5 to 111] | 0.87 [0.72 to 0.94] |
| **THEMINES** | 12 | 0.47 | 11 [9 to 113] | 0.87 [0.69 to 0.96] |
| **THEMINETTES** | 11 | 0.5 | 9.5 [8 to 10] | 0.89 [0.84 to 0.93] |
| **FONTBELLE** | **Global CCA** | | **Median and [Min–Max] of Sliding Windows CCA** | |
| | Lagmax (h) | correlation (Rxy) | Lagmax (h) | correlation (Rxy) |
| **ALZOU** | 6 | 0.43 | 6 [0 to 14] | 0.74 [0.06 to 0.89] |
| **THEMINES** | 10 | 0.48 | 8 [0 to 12] | 0.75 [0.14 to 0.93] |
| **THEMINETTES** | 9 | 0.54 | 8 [0 to 11] | 0.69 [0.028 to 0.93] |
| **ST SAUVEUR** | **Global CCA** | | **Median and [Min–Max] of Sliding Windows CCA** | |
| | Lagmax (h) | correlation (Rxy) | Lagmax (h) | correlation (Rxy) |
| **ALZOU** | 6 | 0.89 | 5.5 [0 to 113] | 0.87 [0.47 to 0.96] |
| **THEMINES** | 8 | 0.87 | 7.5 [1 to 11] | 0.89 [0.24 to 0.96] |
| **THEMINETTES** | 7 | 0.89 | 7 [5 to 19] | 0.91 [0.61 to 0.97] |
| **OUYSSE** | **Global CCA** | | **Median and [Min–Max] of Sliding Windows CCA** | |
| | Lagmax (h) | correlation (Rxy) | Lagmax (h) | correlation (Rxy) |
| **ALZOU** | 8 | 0.85 | 6.5 [1 to 116] | 0.88 [0.1 to 0.98] |
| **THEMINES** | 9 | 0.85 | 9 [0 to 28] | 0.9 [0.74 to 0.98] |
| **THEMINETTES** | 10 | 0.84 | 9 [2 to 28] | 0.85 [0.76 to 0.96] |

This analysis makes it possible to highlight the evolution with time of lag and correlation values for the pressure transfer as a function of the hydrologic regime, particularly during flow events, but also the effect of seasonality on the flow regime.

For all stations, there is a good agreement between the maximum lag (determined from global cross-correlation analysis throughout the entire chronicle) and the median lag (determined from sliding windows cross-correlation analysis) for water level chronicles. All the mean or median lag values estimated between sinks and springs for water elevation are relatively close to each other (varying from 6 to 12 h, regardless of the calculation method). Between the sinks, Alzou seems to have a shorter time lag compared to the Limargue sinks (Themines and Theminettes). In further detail, sliding windows cross-correlation

analysis shows a large variability for lag values that appear to be seasonal: the shorter lags between water level elevation in sinks and outlets are observed in Spring. The larger lag variability is observed between the Alzou sink and the Cabouy and Saint-Sauveur springs (from ~0 to ~115 h). Lag values and lag variability are much lower between the Alzou sink and the Fontbelle spring (0–14 h). Water elevation lag between the Alzou spring and the downstream Ouysse river is consistent with lag values observed at the Cabouy and Saint-Sauveur springs. For the Themines and Theminettes sinks, the evolution of the pressure transfer lag with the system outlets during an annual hydrologic cycle is much more moderate than observed for the Alzou sinks (globally 0–28 h), except between the Themines sink and the Cabouy spring (9–113 h).

Correlation factors calculated for water level elevation between sinks and system outlets are always positive and typically higher than 0.74 (considering the sliding widows method), showing that the hydrodynamic behavior between the sinks and the springs are significantly correlated in the Ouysse system. Moreover, cross-correlation factors determined with water levels between Fontbelle and Cabouy springs and system inlets (Alzou, Themines, and Theminettes sinks) show significant differences between the calculation method considered (global analysis versus sliding windows analysis). The correlation factor is about twice as low for the global analysis. For the Fontbelle spring, the correlation factor with each sink determined from the sliding windows method varies significantly along an annual cycle (from about <0.14 to >0.89). For the Cabouy spring, the variation of the correlation factor with time appears much more limited (from ~0.7 to ~0.9). For the Saint-Sauveur spring and the Ouysse river at Calès, the maximum correlation factors are close for both methods.

These results from cross-correlation analysis of water level between sinks and outlets bring some information on the organization and the hydrodynamic behavior of the karstic catchment:

- In most cases (apart from Theminettes and Cabouy), the delay between water elevation at sinks and water elevation at springs during flood events varies inside a hydrological annual cycle. The lag is shorter in Spring, when the rainfall is maximum (in frequency and intensity) and when the maximum storage capacity of the reservoir is reached. The flood response is thus shorter and more intense during Spring or Summer storms, longer and buffered during the Winter recharge period. Pressure waves have thus the fastest transfer velocity during these high-flow periods, with a minimal loss of amplitude.
- Seasonal lag value variability is much more pronounced in the oriental allogenic sub-catchment of the Ouysse system (mainly between the Themines and Theminettes sinks and the Cabouy spring, and at a minor degree with the Saint-Sauveur spring), than in the occidental autogenic sub-catchment (mainly drained by the Saint-Sauveur and Fontbelle spring). In this latter sub-catchment, the intensity and the reaction time of water level during flood events are quite homogeneous all year long, and much more decorrelated with the water level variations at Themines and Theminettes. This autogenic subsystem appears much more transmissive, with less storage capacity. The rapidity and the intensity of the flood response seem moderately dependent on both the rainfall intensity and the storage state of the reservoir. This subsystem may be relatively simply organized, with a clear major drainage axis with very few minor branching. Conversely, the oriental allogenic subsystem appears less transmissive, with a higher storage capacity, suggesting a bigger development of the karst network, and/or a complex organization including secondary tributaries/branching of unknown development.
- The hydrological behavior regarding the Alzou sink appears singular: water level lags are shorter between the Alzou sinks and the system outlets, probably because the distances between this sink and the springs are shorter (~10 km) than between the other sinks (Themines, Theminettes) and the system outlets (~20 km).

- Lags and amplitude correlation are remarkably close between Themines and Theminettes sinks, regarding the system outlets. The only significant difference is for the maximum lag between Themines and Cabouy (113 h) which is one order of magnitude higher than the other values calculated from cross-correlation analysis (typically lower than 28 h).
- The hydrological behavior of the downstream Ouysse river appears to be mainly correlated to the hydrological behavior of Cabouy and Saint-Sauveur springs, which brings the major volumetric contributions to the main flow of the system outlet.
- From a methodological point of view, the results exposed above show that the presence of rapid and intense variations of water level in karst systems induces a bias in any hydrological interpretation if cross-correlation analysis is conducted only considering a global approach (correlation calculations conducted throughout the whole chronicle) rather than considering a sliding windows cross-correlation method.

### 4.1.2. Crosscorrelation between Sinks and Springs Electric Conductivity

The same cross-correlation analysis that has been conducted on water level elevation to evaluate pressure waves transfer properties can be conducted on electrical conductivity chronicles between sinks and springs in order to evaluate the mass and solute transfer properties of the karstic system (see Table 3).

**Table 3.** Compared results of the global ACC and the sliding window ACC between sinks' electrical conductivity and springs' electrical conductivity.

| CABOUY | Global CCA | | Median and [Min–Max] of Sliding Windows CCA | |
|---|---|---|---|---|
| | Lagmax (h) | correlation (Rxy) | Lagmax (h) | correlation (Rxy) |
| **ALZOU** | 128 | 0.81 | 127 [121 to 133] | 0.74 [0.62 to 0.86] |
| **THEMINES** | 76 | 0.74 | 97 [15 to 130] | 0.77 [0.58 to 0.84] |
| **THEMINETTES** | 84 | 0.64 | 188 [131 to 272] | 0.63 [0.53 to 0.8] |
| **FONTBELLE** | Global CCA | | Median Sliding Windows CCA | |
| | Lagmax (h) | correlation (Rxy) | Lagmax (h) | correlation (Rxy) |
| **ALZOU** | 489 | 0.85 | 66 [0 to 81] | 0.65 [0.55 to 0.67] |
| **THEMINES** | 500 | 0.12 | 67.5 [0 to 480] | 0.46 [0.02 to 0.74] |
| **THEMINETTES** | 185 | 0.3 | 91 [15 to 158] | 0.3 [0.1 to 0.78] |
| **ST SAUVEUR** | Global CCA | | Median Sliding Windows CCA | |
| | Lagmax (h) | correlation (Rxy) | Lagmax (h) | correlation (Rxy) |
| **ALZOU** | 131 | 0.74 | 5 [0 to 255] | 0.57 [0.28 to 0.76] |
| **THEMINES** | 273 | 0.68 | 249 [0 to 353] | 0.6 [0.36 to 0.81] |
| **THEMINETTES** | 264 | 0.65 | 278.5 [164 to 416] | 0.49 [0.13 to 0.75] |
| **OUYSSE** | Global CCA | | Median Sliding Windows CCA | |
| | Lagmax (h) | correlation (Rxy) | Lagmax (h) | correlation (Rxy) |
| **ALZOU** | 392 | 0.36 | 193 [132 to 288] | 0.15 [0.04 to 0.45] |
| **THEMINES** | 248 | 0.21 | 280 [129 to 472] | 0.18 [0.02 to 0.86] |
| **THEMINETTES** | 259 | 0.05 | 130.5 [68 to 320] | 0.23 [0.16 to 0.85] |

The lag values reported for electrical conductivity (between 70 to 500 h) are never negative, and globally one to two orders of magnitude higher than ones determined for water elevation. This suggests a significant decoupling of pressure and mass (solute) transfers inside the Ouysse system, the mass transfer fronts being systematically delayed compared to the pressure waves. Moreover, the differences in lag values brought by the two calculation methods are quite significant. These differences are not systematically oriented in the same way, and the interrelationships between sinks and springs are complicated to outline from the solute transfer point of view:

- For the Cabouy spring, the maximum correlation estimated from the global method and the median correlation estimated from the sliding window method are consistent. The lags of the water mass transfer estimated by the global method vary from ~80 h (from the Themines and Theminettes sinks) to 128 h (from the Alzou sink). In that case, the solute transfer from the Alzou sink to the system outlets seems to be slower than the one from the Limargue sinks. Considering the sliding window method, the lags from the Alzou sink appear to have very limited variations (121–133 h), compared to the ones from Themines and Theminettes that varies on a larger range (respectively 15–130 h and 131–272 h).
- For the Fontbelle spring, the maximum correlation estimated with the global method and the median correlation estimated with the sliding window method are generally inconsistent. The lags of the conductivity estimated by the global method vary from 185 h (from the Théminettes sink) to about 500 h (Alzou and Thémines). In that case, there is a large difference in lag between each sink and the Fontbelle spring, conversely to the estimation made for the Cabouy spring. That is quite surprising, since the Themines and Theminettes sinks are both located at the Limargue inlet of the oriental allogenic sub-catchment. Considering the sliding window method, the median lags estimated appear globally one order of magnitude lower (from ~67 h from the Alzou and Themines sinks, to 91 h from the Theminettes sink), but with a variability that can be very high (notably in the case of the Themines sink: 0–480 h).
- Comparable observations can be made for the Saint-Sauveur spring and the downstream Ouysse river: maximum lags estimated from both methods can be fairly consistent, but can vary inside a large range (e.g., 0–353 h from Themines to Saint-Sauveur).

The organization of solute transfer between upstream inlets to downstream outlets is thus difficult to interpret with cross-correlation analyses because at least three water bodies of different mineralization mix into the Ouysse system [17], as listed in Table 1:

- The water from the rivers comes from the Limargue area and is fed by Segala igneous bedrock springs, with low mineralization, but relatively high concentrations of $Cl^-$, $Na^+$, and $K^+$.
- The highly mineralized water from the Alzou subsystem is characterized by a high conductivity due to its evaporitic origin.
- Water directly infiltrated on the autogenic karst area, with chemistry controlled by the dissolution of the Middle and Upper Jurassic limestones.

The mixing of these water bodies is realized in various proportions but also with various delays from the sinks to the outlets. The mixing is thus not synchronized between the water bodies during a flood event, leading to a large range of mixing solutions, in terms of water composition and solute transfer delays.

To calibrate these results, a tracer test was carried out between the Alzou river and the outlet springs during the October 2020 flood with an injection of 10 kg of fluorescein dye. The mass transfer time of the dye is approximately 3 days for all outlets (74 h for Cabouy, 76 h for Saint-Sauveur and Fontbelle) (Fluoresceine peaks for each spring are shown in Appendix B, Figures A7–A9).

The dye transfer of about 70/80 h is close to the median time lag of the sliding cross-correlation results between the Thémines and Théminettes sinks and the Fontbelle spring (Table 3). The values are slightly higher at the Cabouy spring for the time lag calculated with

the global CCA and much higher at the Saint-Sauveur spring but as previously mentioned, given the multiplicity of signals observed for one flood at those springs, the CCA methods struggle to isolate the signal corresponding to one specific origin. However, since the tracer tests showed that the Alzou water has a similar mass transfer time between Saint-Sauveur and Fontbelle, the transfer time for the Thémines and Théminettes waters should also be similar for these two springs.

For the mixing model, it will therefore be considered that there is an approximately three-day delay in the transfer of the water mass between all river sinks and all Ouysse springs. For the Courtilles borehole, this time lag will be reduced to 2 days (results of a simple CCA on the flood electrical conductivity results).

### 4.2. Mixing Model Results

With the previously estimated lag in the mass transfer, it is possible to use it in a mobile source mixing model which will be applied to the flood monitored in October 2020.

#### 4.2.1. Geochemical Decomposition of the Flood Chemiograph during High-Frequency Flood Monitoring

The weather station recorded a first rainfall period of 95 mm cumulated from 1 October 2020 at 10 h to 6 October 2020 at 22:00. Until 13 October 2020, no rainfall over 1 mm/h is recorded, then a second rainfall period of 19 mm is observed from 13 October at 2:00 to 14 October at 22:00. No other rainfall is measured until the end of the high-frequency sampling period (23 October at 7:00).

Table 4 provides a summary of the sampling results, full graphical results for each station are shown in Appendix B.

**Table 4.** Summary of statistics on the flow/water level, temperature, and conductivity at each monitored station.

| | | Sinks | | Borehole | | Springs | |
|---|---|---|---|---|---|---|---|
| | | **Themines** | **Alzou** | **Courtilles** | **Cabouy** | **Saint Sauveur** | **Fontbelle** |
| Ca (mg/L) | min | 29.72 | 125.65 | 58.61 | 82.77 | 99.95 | 116.00 |
| | median | 42.97 | 217.00 | 86.38 | 124.37 | 126.00 | 131.58 |
| | max | 82.20 | 446.00 | 114.10 | 153.99 | 151.00 | 172.78 |
| | variance | 5.96 | 268.51 | 10.41 | 12.54 | 8.31 | 4.54 |
| $HCO_3$ (mg/L) | min | 106.14 | 200.08 | 173.24 | 302.56 | 300.00 | 306.22 |
| | median | 152.50 | 275.72 | 259.86 | 335.50 | 340.38 | 337.33 |
| | max | 263.52 | 329.40 | 322.08 | 417.24 | 531.79 | 451.40 |
| | variance | 28.55 | 18.65 | 19.62 | 17.24 | 25.54 | 8.42 |
| $SO_4$ (mg/L) | min | 13.10 | 127.00 | 11.70 | 8.56 | 2.53 | 4.32 |
| | median | 20.00 | 400.00 | 15.50 | 16.00 | 8.58 | 7.25 |
| | max | 71.70 | 921.00 | 44.30 | 73.70 | 70.60 | 64.40 |
| | variance | 3.92 | 853.66 | 1.34 | 7.84 | 6.50 | 3.53 |
| K (mg/L) | min | 3.27 | 2.96 | 1.35 | 0.38 | 1.08 | 0.50 |
| | median | 4.22 | 5.41 | 2.70 | 1.99 | 2.11 | 1.23 |
| | max | 11.63 | 8.63 | 4.02 | 3.77 | 5.01 | 2.76 |
| | variance | 0.10 | 0.07 | 0.02 | 0.01 | 0.01 | 0.01 |

**Table 4.** *Cont.*

|  |  | Sinks | | Borehole | | Springs | |
|---|---|---|---|---|---|---|---|
|  |  | Themines | Alzou | Courtilles | Cabouy | Saint Sauveur | Fontbelle |
| Cl (mg/L) | min | 8.87 | 7.85 | 7.95 | 7.95 | 2.43 | 5.68 |
|  | median | 10.40 | 12.20 | 8.83 | 8.84 | 7.99 | 8.38 |
|  | max | 14.90 | 18.00 | 9.87 | 14.70 | 12.20 | 10.40 |
|  | variance | 0.03 | 0.23 | 0.01 | 0.09 | 0.14 | 0.10 |
| Na (mg/L) | min | 4.41 | 2.42 | 2.39 | 0.46 | 2.78 | 0.65 |
|  | median | 6.17 | 6.16 | 4.63 | 3.57 | 3.86 | 3.55 |
|  | max | 11.22 | 9.01 | 6.82 | 7.83 | 7.54 | 6.17 |
|  | variance | 0.10 | 0.12 | 0.04 | 0.15 | 0.09 | 0.06 |
| Mg (mg/L) | min | 8.26 | 11.50 | 6.89 | 1.94 | 2.55 | 0.81 |
|  | median | 12.27 | 24.55 | 11.03 | 4.70 | 3.95 | 3.22 |
|  | max | 20.20 | 59.60 | 14.97 | 9.09 | 10.60 | 8.21 |
|  | variance | 0.67 | 9.82 | 0.41 | 0.25 | 0.29 | 0.15 |
| NO$_3$ (mg/L) | min | 8.01 | 0.04 | 10.90 | 5.39 | 5.14 | 422.84 |
|  | median | 12.20 | 22.30 | 16.10 | 19.55 | 11.90 | 13.40 |
|  | max | 13.80 | 52.20 | 17.80 | 26.20 | 30.80 | 30.90 |
|  | variance | 0.05 | 4.44 | 0.06 | 0.49 | 1.03 | 1.45 |

All the connections between the outlets, the karst, and the springs can be observed through the different chemical inputs of the source water bodies. It is possible, and desirable, to go further in the analysis and quantify these different inputs in mixed samples at the outlets and intermediate stations [18–22]. Such an analysis has already been performed for the Ouysse karst system on a 2016 flood [17]. This analysis was reproduced here on a 2020 flood for two reasons. First, the study was only focused on monitoring the Courtilles borehole and the Cabouy spring; it, therefore, seemed interesting to extend the method to the Fontbelle and Saint-Sauveur springs. Secondly, the Alzou sink had not been monitored either, so it was not possible to consider the real-time evolution of all the inputs. These two issues were addressed during the 2020 flood monitoring to achieve a more complete mixing model.

4.2.2. Source Water Bodies Signatures Identification

To quantify the chemical contributions of each origin (the Alzou sink (1), the Themines/Theminettes Limargue sinks (2), and the autogenic karst (3)) in the hydrochemical signature of the Courtilles borehole and the Cabouy spring, it is essential to choose the most discriminating variables as parameters of the Equation (2). It was observed that dilution caused by rainfall events had very little influence on the chemical mixing in previous work [17].

To select these variables, a principal component analysis (PCA) was performed in previous work [17] with major ions concentrations sampled during the May 2016 Flood at the Francès sink at Théminettes, the Courtilles borehole, the Cabouy spring, and the Combettes underground river.

The variables selected as the most discriminating in this previous work are: (i) HCO$_3^-$ for the autogenous karst; (ii) SO$_4^{2-}$ as the marker for the Alzou pole; (iii) K$^+$ for the Limargue sinks.

Considering the high variance in all origins in these three parameters (Table 4), it seems preferable to include all variations of each pole in a mobile source mixing model rather than having a fixed sample for each origin in the mixing model. In the autogenic karst, the variability is mainly linked to a difference in speed between "young" water (not very mineralized, it is rainwater that crosses the karst and interacts little with the rock) and "old" water (water stored in the karst and remobilized by the piston effect, which has been equilibrated with the rock for a long time, and which is very mineralized). These variations are not very wide and sampling in underground rivers on a daily frequency for several days implies costly and risky operations. This end member will therefore be considered as fixed. This point is the main weakness of the model.

As mentioned in the previous chapter with CCA and tracer tests, a time lag of 3 days between the sinks and the springs is selected. The first samples considered by the model are those of 26 September 2020, the date of the first sample at the sinks, and 29 October 2020 for the springs (3 days later). A time lag of two days is used for the Courtilles Borehole which is located closer to the sinks. The first sample at Courtilles was on 3 October 2020 (2 days of time lag after the second sample of the sinks dated 1 October). The flood pressure wave transfer already arrived at the springs on these dates, but the water mass and solute transfer were on the way as attested by the tracer test.

The respective contributions of the two origins feeding the Courtilles borehole (the autogenic karst and the Limargue sinks) can be calculated for each flood sample using the concentrations of the variables $HCO_3^-$ (karst), and $K^+$ (Limargue) in Equation (2). The results are shown in Figure 5B.

At the beginning of the flood on the first measurement point, the shares of water coming from the karst and the Limargue are very contrasted, with the contribution of the Limargue sinks calculated at 100%. Then, during the flood, the contributions are balanced and then reversed from 5 to 12 October, with up to 70% of the water coming from the autogenic karst and 30% from all sinks. On 15 and 16 October, the balance returns with 50% of each origin. On 16 October, a new flood arrives, and the contributions of the karst go up to 70%. Then the level goes down again, and at the last sample (21 October) a tendency to return to the initial state is observed with a 75% contribution of the sinks against only 25% of the autogenic karst.

This shows that at low water levels in the underground river of Vitarelles, the majority or even the totality of the chemical contributions comes from the sinks of Limargue whereas during the floods the autogenic karst takes over and becomes the major contributor. These observations are different from what was calculated in the previous deconvolution [17]. This is likely related to the difference in the initial states for the two floods studied: one was in May 2016 and was therefore one of the last floods before low water, and the karst was not yet drained. The October 2020 flood is the first big flood of the hydrological cycle and therefore arrives just after the low water level, the karst levels are at their lowest and the permanent river sinks, therefore allowing to maintain the base level. This is why most of the water comes from the Limargue in 2020, while in 2016, water mostly comes from the karst. Then, once the rainwater feeds both the autogenic karst and the sinks, the mix becomes homogenized, and equivalent contributions are observed from the karst and the sinks for both the 2016 and 2020 floods.

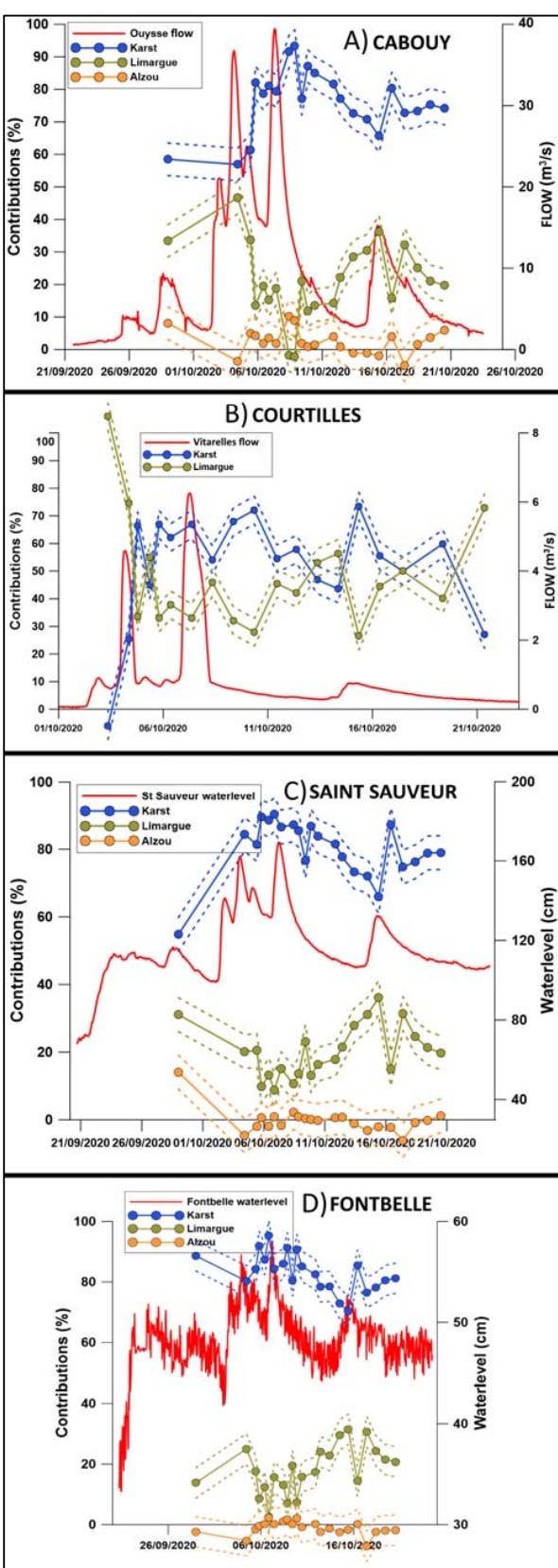

**Figure 5.** Evolution of the contributions results of the mobile source mixing model for Cabouy (**A**), Courtilles (**B**) Saint Sauveur (**C**), and Fontbelle (**D**) alongside water level. The dotted lines correspond to a positive and negative 5% error for each point.

The hydrochemical decomposition at Cabouy is performed with the variables $HCO_3^-$, $K^+$, $SO_4^{2-}$ in Equation (2), marking the waters of the autogenic karst, the Limargue sinks, and the Alzou respectively. Figure 5A illustrates these results. Proportions are stable before the flood's arrival, with the autogenous karst as the main contributor (~60%) and with complementary contributions from the Alzou (~10%) and Limargue (~30%) sinks. During the two flood peaks, the karst water contributes up to 90% while the contributions of the Limargue sinks drop to less than 10%. The contribution of the Alzou sink becomes negligible (~0%). However, during the first peak, we observe a slight increase in the contribution of the Limargue (~50%) which could be linked to the fact that we observe the first flood after the low water level. This increase is not observed at the second peak nor on the decomposition of the 2016 flood because the karst and epikarst are filled by the last flood and the piston effect counterbalances the arrival of the loading of the sinks. After the flood, the proportions of water from each origin tend to return to the initial values (karst ~75%, Limargue ~20%, Alzou ~5%). The contributions are quite similar to what was observed in the 2016 flood deconvolution (Lorette et al., 2020) with, however, smaller contributions from the Alzou (30–10%) in low water and larger contributions for the Limargue sinks (30% in low water). In both cases, the karst/sink proportions are very similar.

For this 2020 flood, the Saint-Sauveur and Fontbelle springs were also monitored and deconvolution with the same poles as Cabouy was performed (autogenic karst with $HCO_3^-$, Limargue sinks with $K^+$, and Alzou with $SO_4^{2-}$).

For Saint-Sauveur (Figure 5C), the pre-flood contributions are similar to Cabouy, with a dominance of autogenous karst (~55%) and complementary contributions from the Alzou (~15%) and Limargue (~30%) sinks. During the two flood peaks, the karst water contributes up to 90% while the contribution of the Limargue sinks drops to less than 10%. The contribution of the Alzou sinks becomes negligible (~0%). After the flood, the values seem to take longer to recover their initial values (karst ~80%, Limargue ~20%, Alzou ~0%).

For Fontbelle (Figure 5D), water from the Alzou sink does not appear to contribute to its chemistry in any of the samples. The model results are consistent with the results of the tracer test done at the Alzou which showed negligible restitution at Fontbelle. The influence of the flood is much lower than at the other springs, but there is a slight increase in the contributions of the karst during the peak of the flood, from 80 to 90%. The remaining contributions are attributed to the Limargue sinks (between 10 and 20%). As for Cabouy, there is a slight increase in the contributions of the Limargue at the beginning of the first flood peak, probably related to the fact that the flood follows a prolonged low water period.

A synthesis of these results was compiled in the conceptual model illustrated in Figure 6.

These results lead to a better understanding of how the Ouysse karst system works. The fact that the water mass transfer at Cabouy spring was significantly influenced by the river sinks and the Saint Sauveur and especially the Fontbelle springs were mainly fed by autogenic karst was already understood with the tracer tests results, but the mixing model results allowed to quantify in which proportion each end member was represented at each outlet. It also allowed us to understand the evolution of these proportions according to the aquifer hydrodynamic variations.

This latter point will be especially useful for the management of the drinking water resource at Cabouy. This spring water is pumped for the Rocamadour medieval city in which its small population of 600 inhabitants sees up to 1.5 million visitors, mainly in Summer. The water supply demand is thus much higher in this period, and this study shows that during low flow, the Cabouy spring is much more influenced by river sinks. Particular caution must thus be applied in the monitoring of the water quality of those sinking rivers in summer.

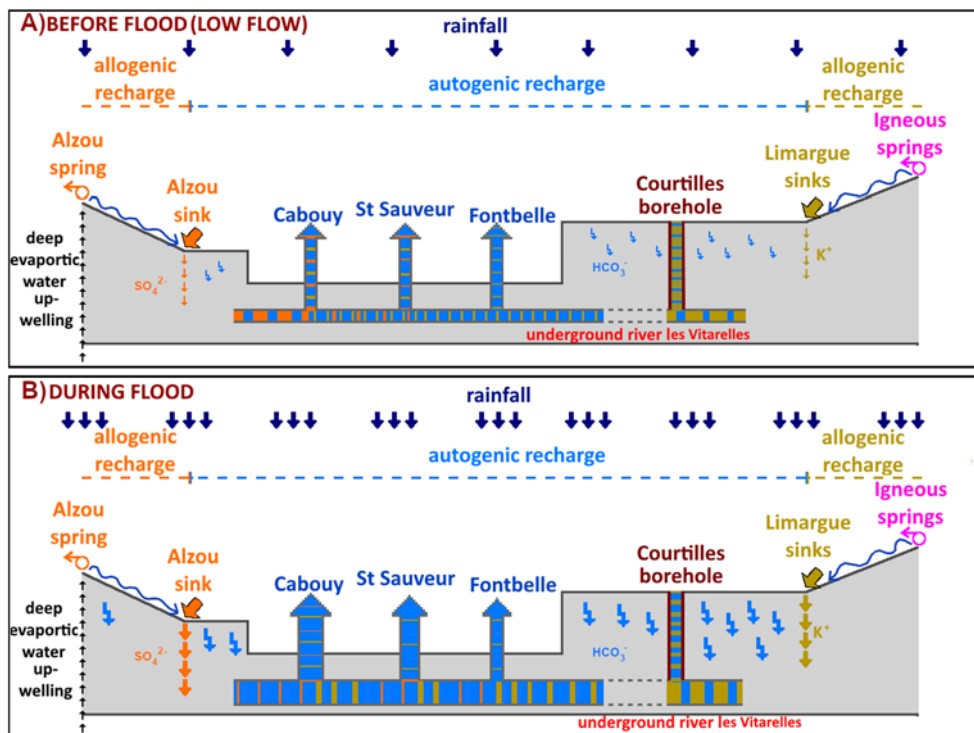

**Figure 6.** Conceptual model synthesizing the results of the mixing model.

These mobile source mixing model results can be compared to what results could be obtained if a fixed source model was used. The fixed source model was described in previous work [17]. The fixed values were determined there as the maximum values of each pole. The variations of the contributions calculated with both fixed and mobile models were then compared to see if any significant differences can be observed. The contribution curves were compared in Table 5 using the NASH and the correlation index between each method's results. Significant differences are observed for Limargue contributions at Cabouy, Saint Sauveur, and Fontbelle (both index < 0.7). Karst's contributions show differences between Cabouy and Fontbelle. Alzou's contributions also show clear differences for Fontbelle.

**Table 5.** Comparison between the contribution curves of the mobile and fixed origin mixing model.

|  | NASH | | | Correlation | | |
|---|---|---|---|---|---|---|
|  | **Limargue** | **Karst** | **Alzou** | **Limargue** | **Karst** | **Alzou** |
| cabouy | 0.04 | 0.17 | 0.89 | 0.19 | 0.41 | 0.94 |
| saint sauveur | 0.24 | 0.70 | 0.76 | 0.49 | 0.84 | 0.87 |
| fontbelle | 0.44 | 0.52 | 0.21 | 0.66 | 0.72 | 0.45 |

Figures 7–9 allows looking at those differences in detail by showing the contributions curves for both mobile and fixed origin mixing model. In all cases, the biggest discrepancies are observed for the first three samples (29 September, 5 and 6 October).

During the flood, the fixed sample contributions are well included in the 5% error range of the mobile model and no significant changes are observed at any outlet for any end member. After 12 October, discrepancies are observed again. These differences concentrate on the Limargue and Alzou contributions with wider variations for the mobile origin model compared to the fixed origin model, which shows a more linear evolution for Limargue and Alzou contributions at all springs.

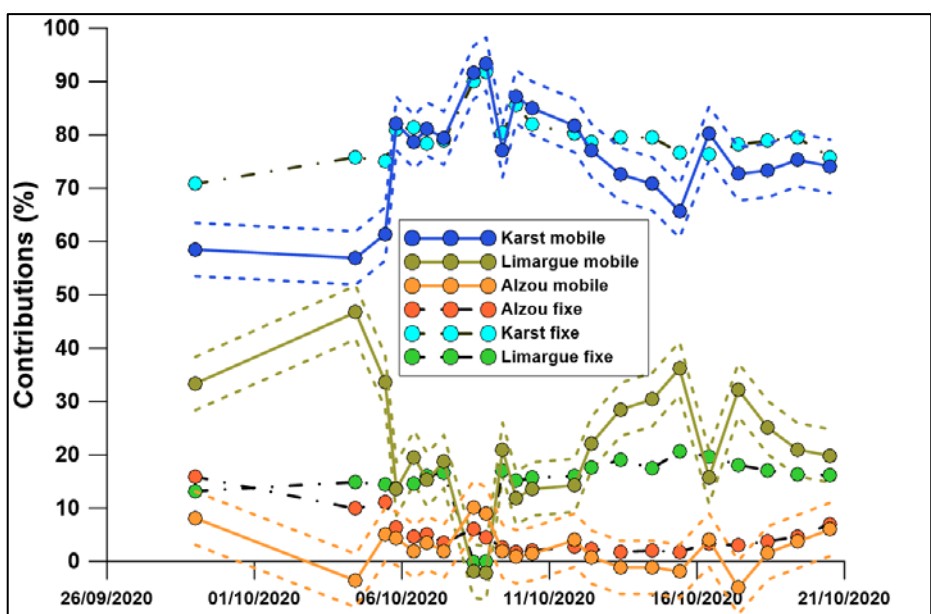

**Figure 7.** Contributions results of the mobile source mixing model for Cabouy compared with fixed source model.

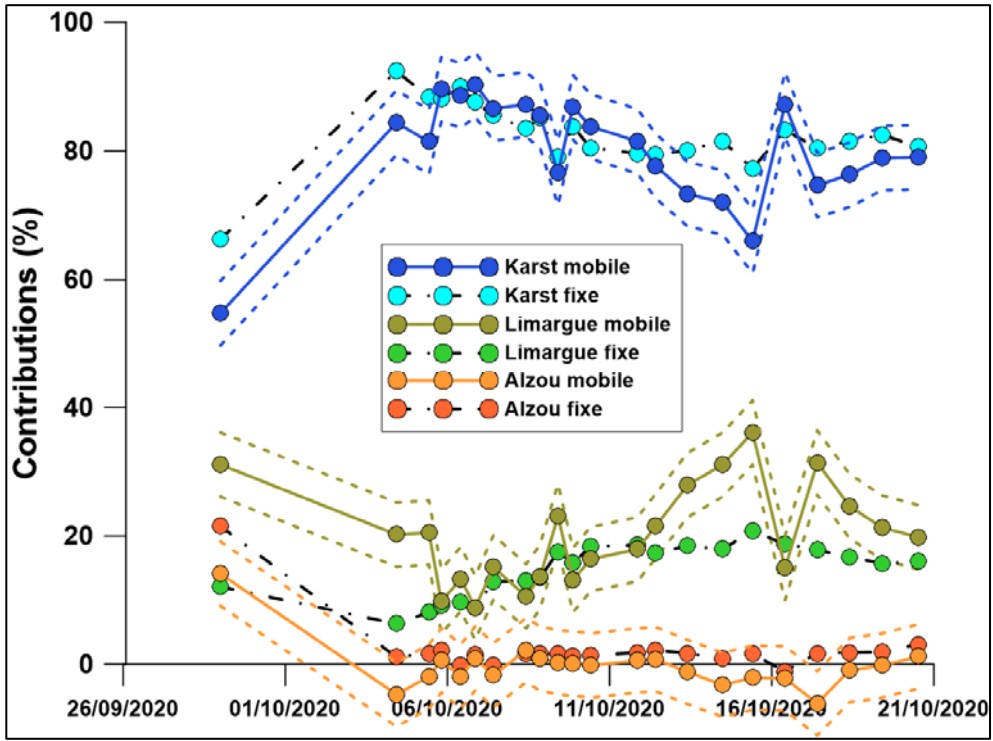

**Figure 8.** Contributions results of the mobile source mixing model for Saint Sauveur compared with fixed source model.

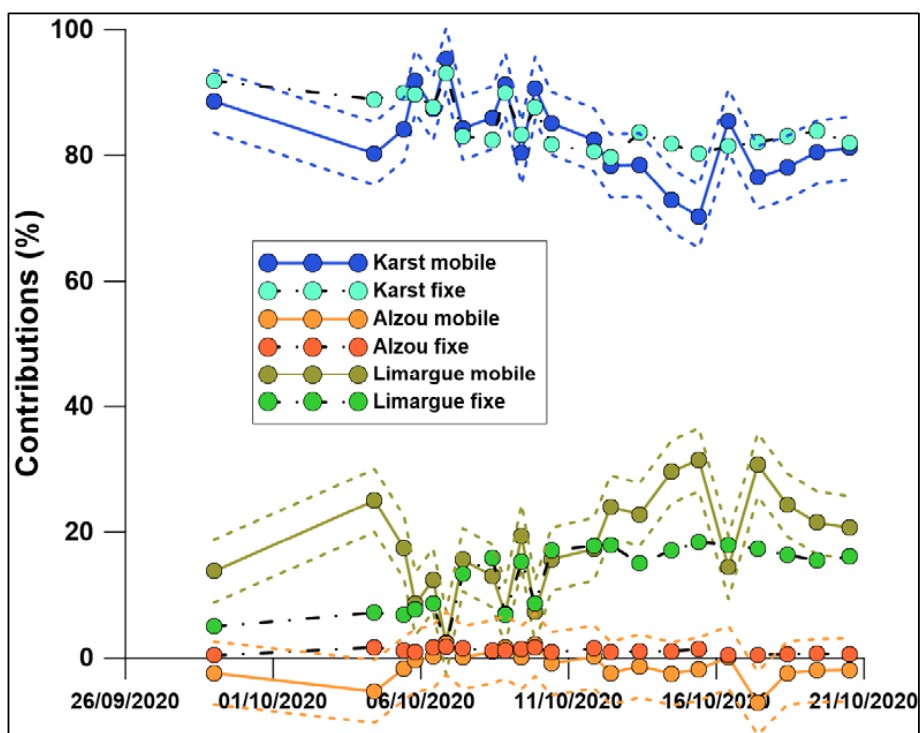

**Figure 9.** Contributions results of the mobile source mixing model for Fontbelle compared with fixed source model.

The mobile origin model is based on an approach that is closer to what happens physically in the aquifer and seems more sensitive to variation, whereas the fixed origin model is easier to carry out (no need to monitor each end member with high frequency, only the outlets), and its results give tendencies similar to the mobile origin model.

In the case of the autogenic karstic system showing a very high temporal variability like the Ouysse karst system, taking variations of each source in a mobile origin mixing model seems to be necessary to have a more realistic model, but a fixed origin mixing model can still be an interesting solution as a more economical solution.

## 5. Conclusions

Water management in a karst environment requires a good knowledge of water origins and feeding the outlets used as drinking water resources. In other words, the quantification of water sources is a major step for remediation measures to protect the resource. This work highlights the way to implement an innovative method of mobile source mixing model to help quantify hydrochemical sources in a mixed karst system. The results showed the use of this innovative approach in the case of a karst system with mixed origins showing a very high temporal variability, a mobile origin mixing model being more sensitive to variation and more realistic.

From a fundamental perspective, this study proposes a new range of methodology applicable to the study of multiple origin catchments. Future steps in this research will be to adapt this method to quantify water origins at the scale of hydrological cycles. For the Ouysse karst management, the research is oriented towards (i) hydrological modeling and (ii) water quality perturbation focusing on nitrate transports from several water origins using nitrate isotopes. All these studies are integrated into global water management over the Causse du Quercy UNESCO Global Geopark.

**Author Contributions:** Conceptualization: D.V., G.L., D.L., M.F., M.S. and P.C.; methodology: D.V., G.L., D.L., M.F., M.S. and P.C.; software: D.V., G.L.; validation: G.L., D.L., M.F., M.S. and P.C.; formal analysis: D.V.; investigation: D.V., G.L. and O.A.; resources: D.V., G.L. and O.A.; data curation: D.V.,

G.L., P.C. and O.A.; writing—original draft preparation: D.V.; writing—review and editing: D.V., G.L., D.L., M.F., M.S. and P.C.; visualization: D.V.; supervision: D.V., G.L. and P.C.; project administration: D.V., G.L. and P.C.; funding acquisition: D.V. and G.L. All authors have read and agreed to the published version of the manuscript.

**Funding:** This research was funded by the Région Occitanie, the Adour-Garonne Water-Agency, the Conseil Départemental du Lot, and the French Alternative Energies and Atomic Energy Commission (CEA).

**Data Availability Statement:** Data is available at https://data.oreme.org/observation/snokarst, accessed on 16 December 2022.

**Acknowledgments:** The authors thank the speleologists who contributed to the installation and maintenance of the high-resolution monitoring probes in the underground rivers. They also thank Cyril Delporte and André Tarrisse for their hydrogeological knowledge of the Ouysse karst system. This work benefited from fruitful discussions within the framework of the KARST observatory network (www.sokarst.org) initiative from the INSU/CNRS, which aims to strengthen knowledge-sharing and promote cross-disciplinary research on karst systems.

**Conflicts of Interest:** P.C. and O.A. works for the CEA (French Alternative Energies and Atomic Energy Commission) which is involved in the funding of this research.

## Appendix A

Sliding window cross-correlation detailed figures

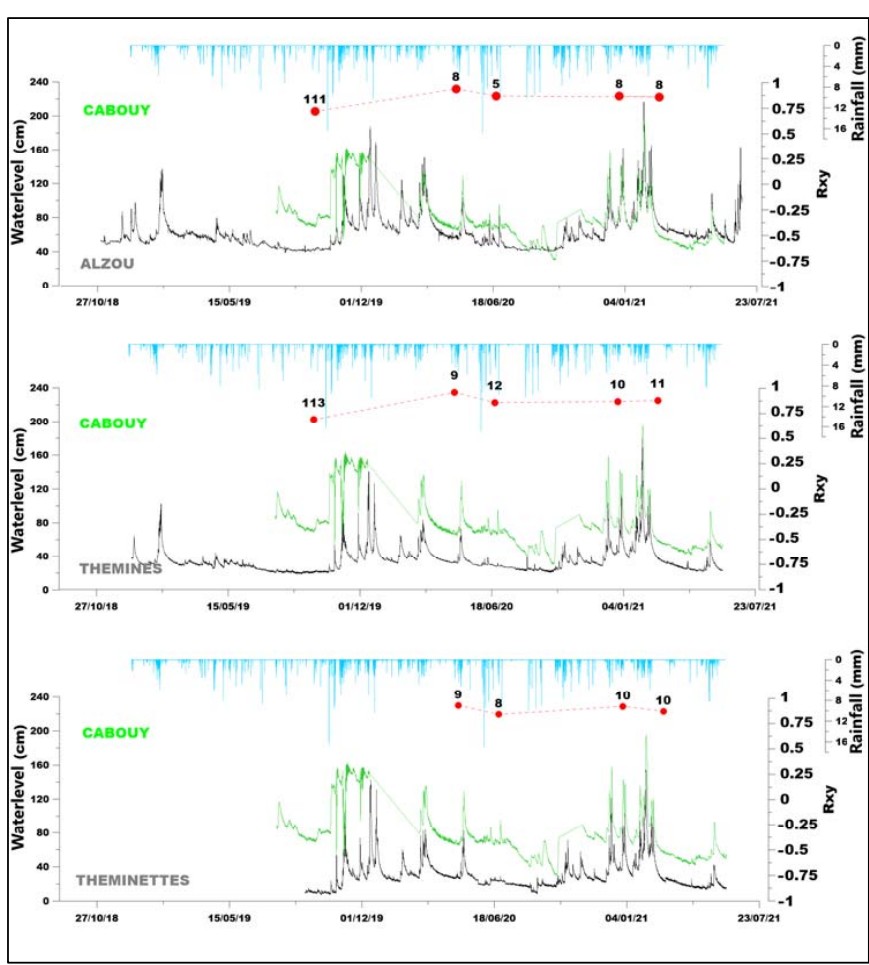

**Figure A1.** Sliding correlation analyses between the water level for the Cabouy station and the different stations at the sinks; the results of the sliding cross-correlation analyses are displayed in red dots (vertical axis: correlation, label: time lag between the two sites in hours).

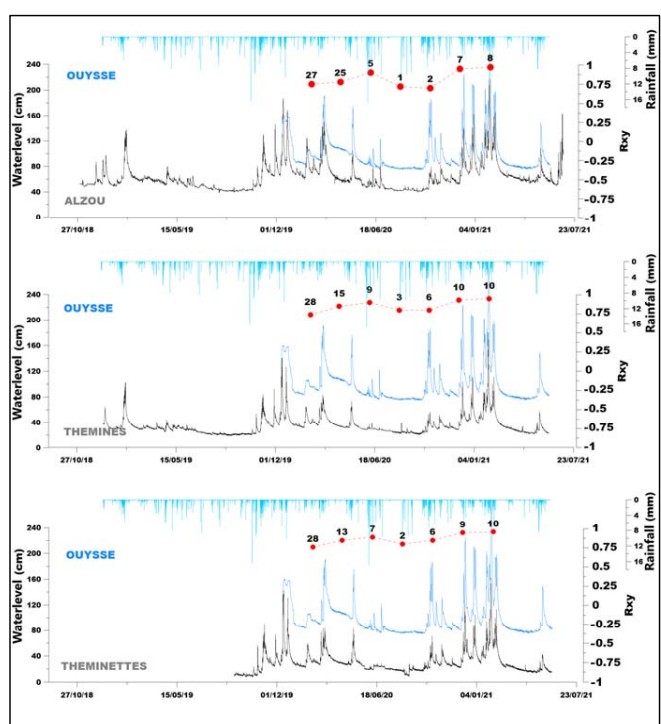

**Figure A2.** Sliding correlation analyses between the water level for the Ouysse station and the different stations at the sinks; the results of the sliding cross-correlation analyses are displayed in red dots (vertical axis: correlation, label: time lag between the two sites in hours).

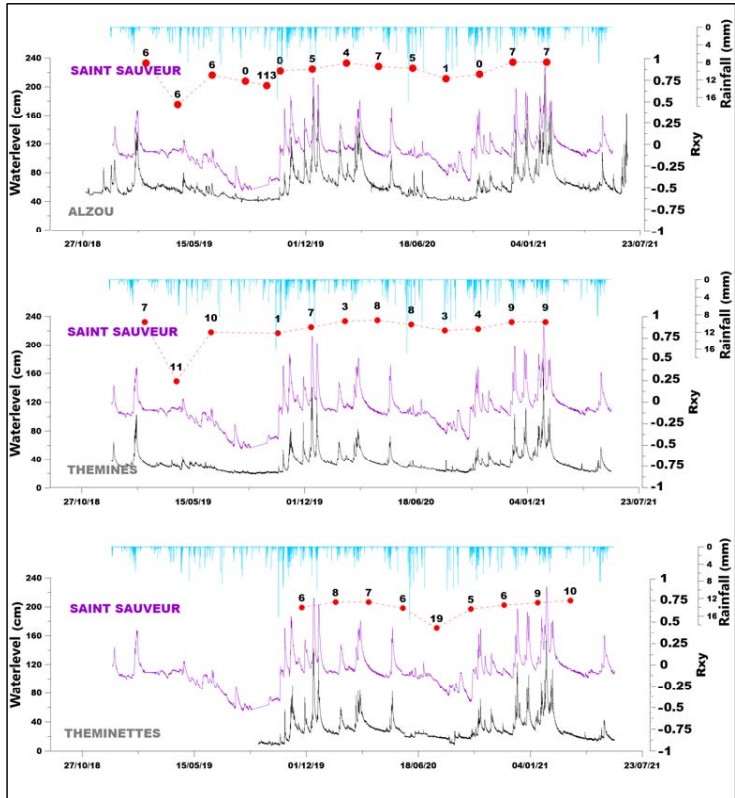

**Figure A3.** Sliding correlation analyses between the water level for the Saint Sauveur station and the different stations at the sinks; the results of the sliding cross-correlation analyses are displayed in red dots (vertical axis: correlation, label: time lag between the two sites in hours).

## Appendix B

Detailed chemiographs of the October 2020 flood

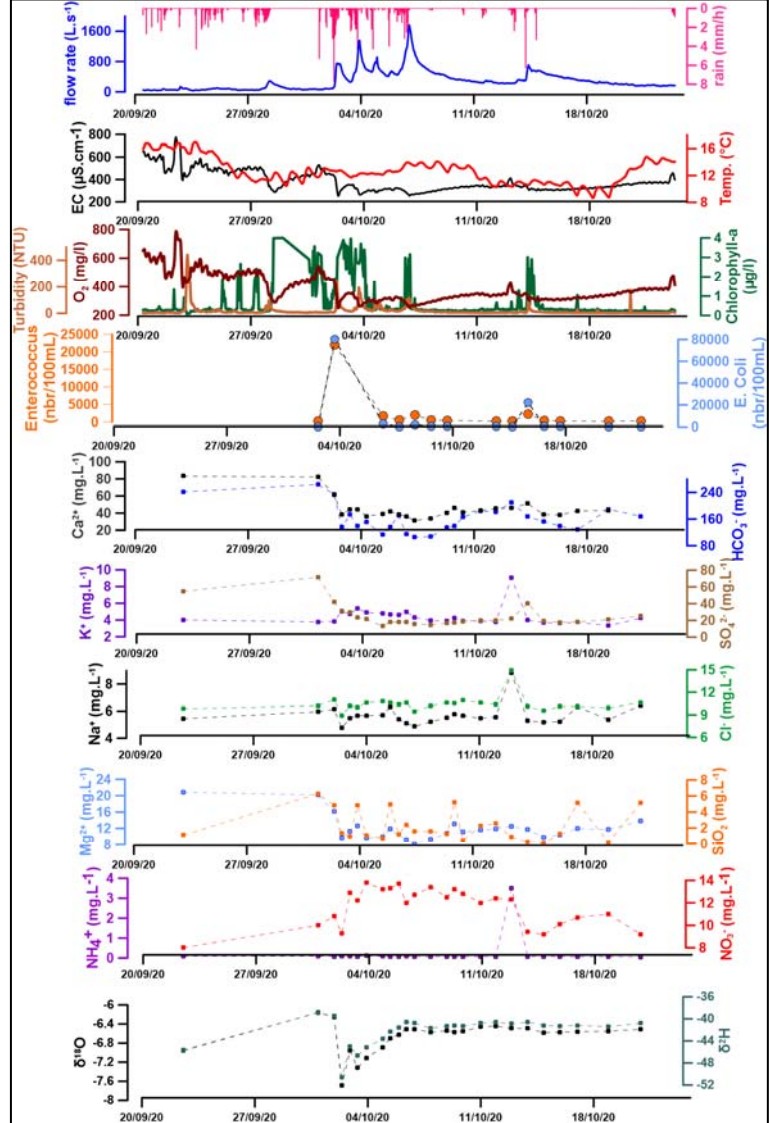

**Figure A4.** Chemiograph of the Themines sink from 20/09/20 to 21/10/20.

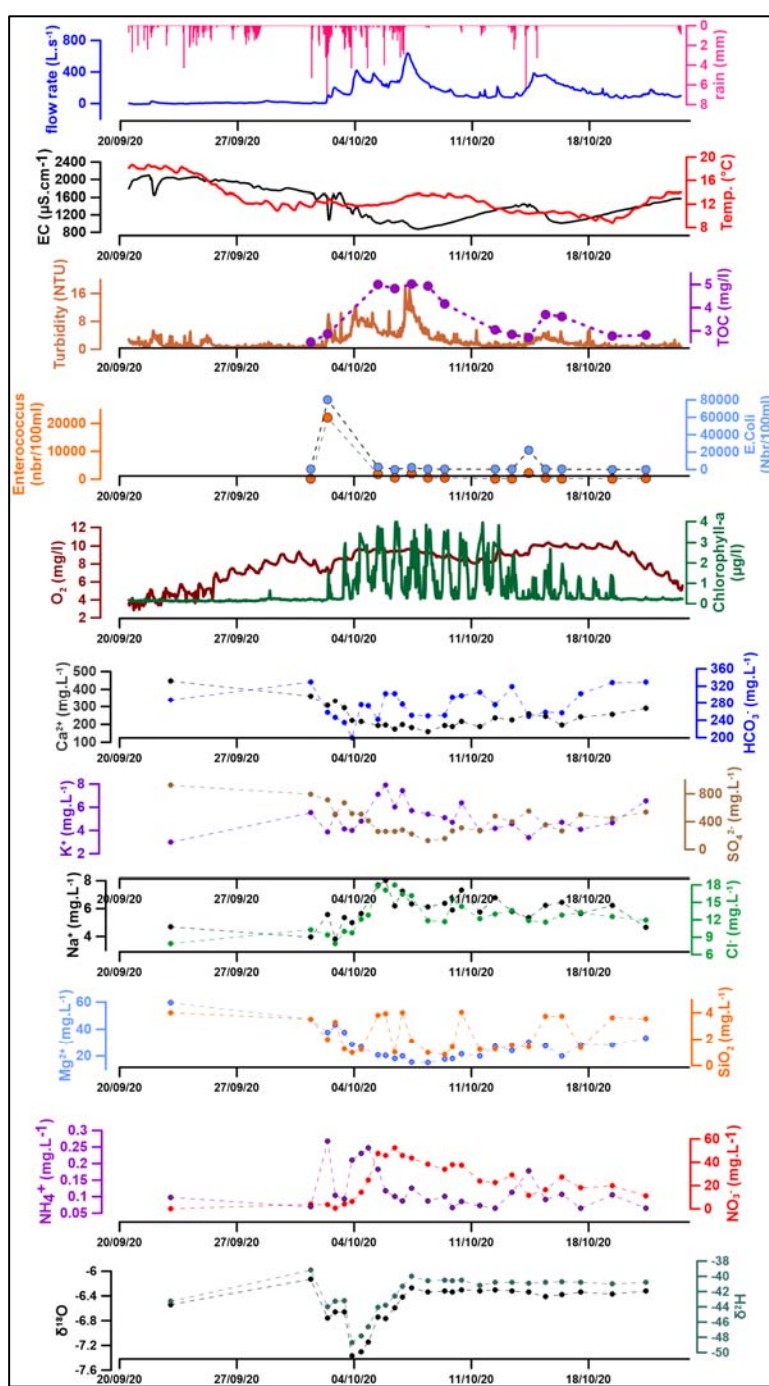

**Figure A5.** Chemiograph of the Azlou sink from 20/09/20 to 21/10/20.

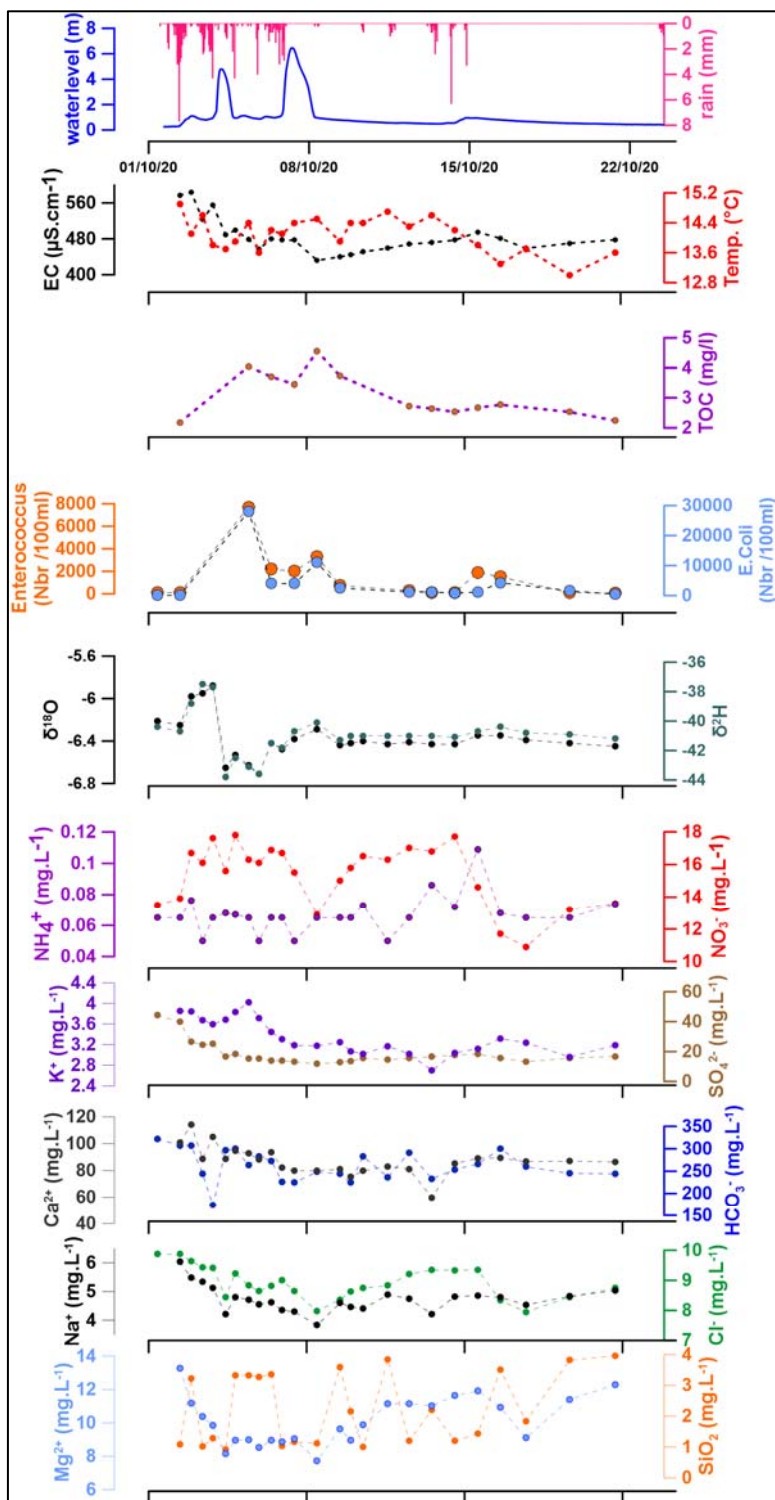

**Figure A6.** Chemiograph of the Courtilles borehole from 20/09/20 to 21/10/20.

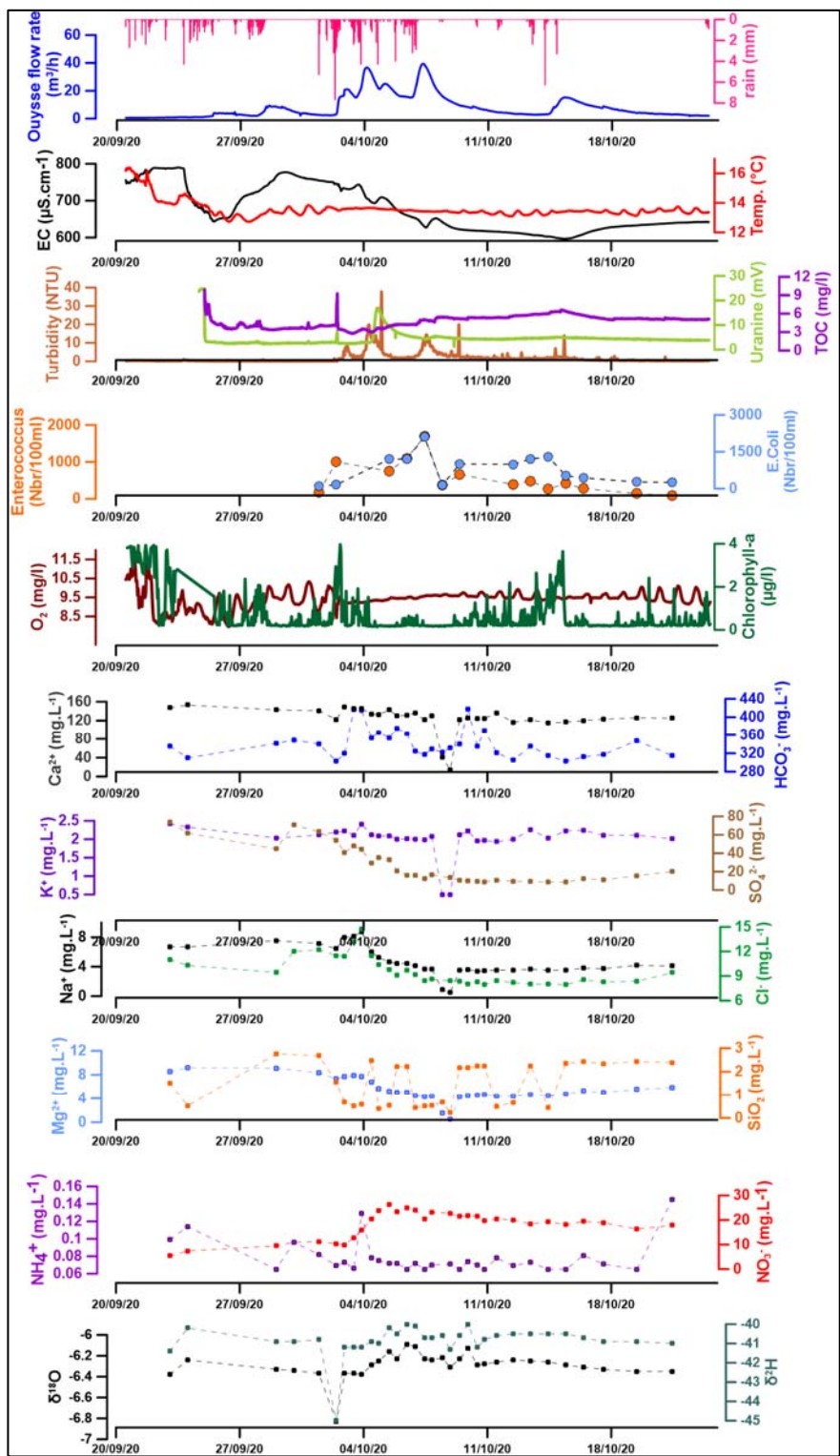

**Figure A7.** Chemiograph of the Cabouy spring from 20/09/20 to 21/10/20.

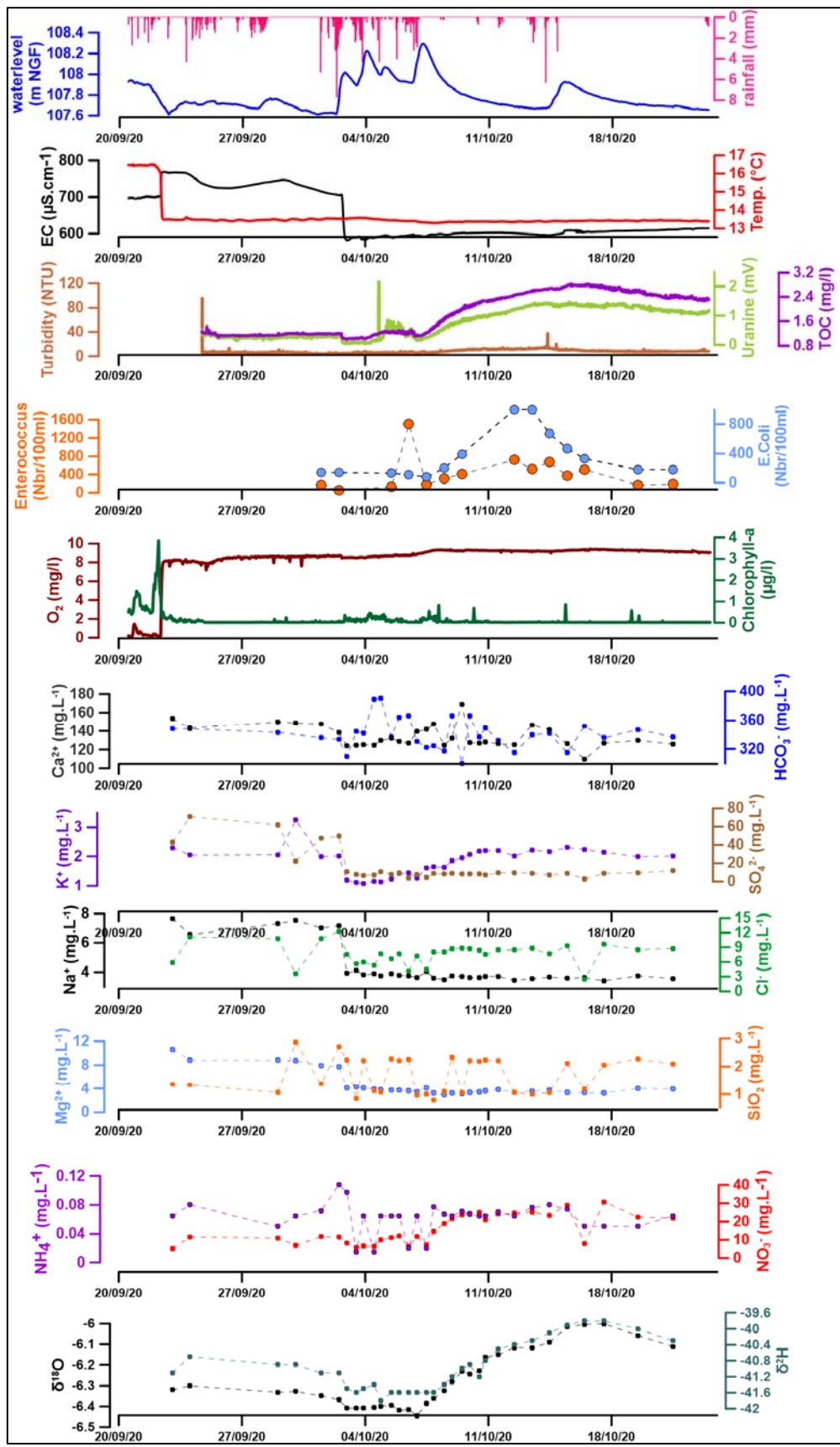

**Figure A8.** Chemiograph of the Saint Sauveur spring from 20/09/20 to 21/10/20.

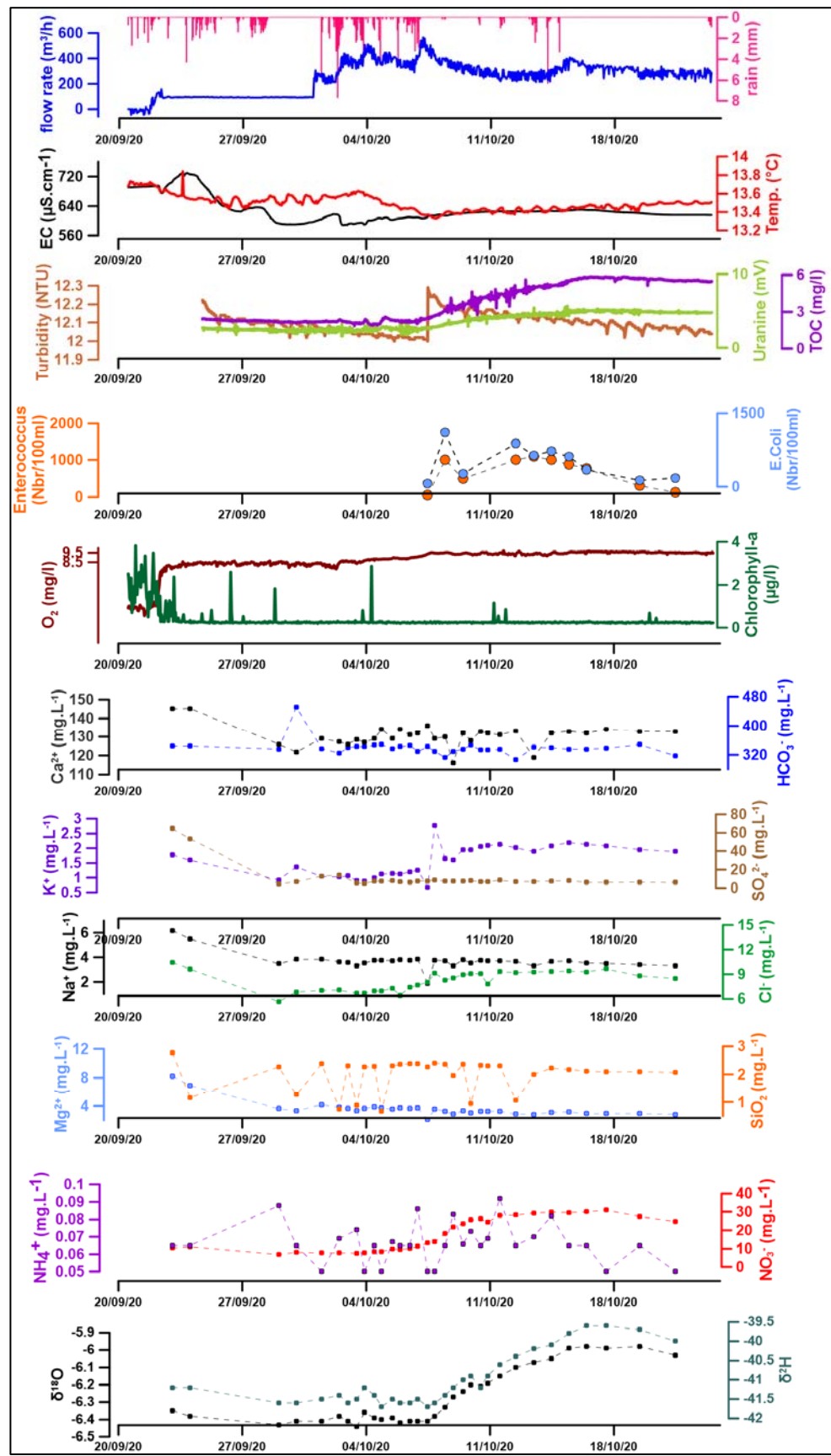

**Figure A9.** Chemiograph of the Fontbelle spring from 20/09/20 to 21/10/20.

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
