# Peer review of "Mobile Sources Mixing Model Implementation for a Better Quantification of Hydrochemical Origins in Allogenic Karst Outlets: Application on the Ouysse Karst System"

_water, doi:10.3390/w15030397_

Round 1

Reviewer 1 Report

The authogenic and allogenic karst aquifer recharge from rainfall. The stream, springs, and aquifer are hydrogeologically interconnected and recharge each other.

The paper was very important especially in droughts periods. They harvest the surface water or inject it for dry season. There is some comments should kept in mind:

1 the topography of the karts landscape should illustrate here, because they influence on the recharge - discharge rate

2 the permeability of the limestone (fractures and joints) must illustrated to confirm these contributions from different sources.

3 equations 1 and 2 for the concentration ignored completely the rock water interaction between limestone (CaCO3) and aquifer (groundwater).

 4 the major ions units in table 4 and appendices, I thought, was different, check

5 the drainage streams also influence on sinks and contributions, you ignored them

6 Al Alzou sinks, there is confliction in interpretation about the aquifer recharge, please look carefully in attached manuscript comments.

7 The authors said the precipitation contributed very low contribution into the aquifer system, which opposed with the Fig. 4.

8 the major ions in karst limestone (CaCO3) were low concentration, it needs explanation, why.

9 The contribution of different water sources into sinks was affected by geological, hydrogeological, and hydrological characteristics. The authors ignored them completely.

10 I thought the rainfall chemistry in this region was very important. You can compare it and the other water sources. The hydrogeochemical changes of rainwater when infiltrated into karst aquifer should be considered.

11 Correct the comments contained in attached manuscript

Reviewer 2 Report

The study is good

- Text is too long and classic study in these karst aquifer

- Try to add some values about the salinity, facies, resources... of water in the abstract

- In the introduction, try to more cite some other areas of karst aquifer in the word (Tunisia-Ayadi et al 2018), in Algeria (Hamad et al 2018), in USA-San antonio county (USGS)...

- Added the geographical coordinates to the figures 1

- Page 8, the piston flow, verify this processus with the duration of humide period in your study area (liquid rainwater and/or ice). Don't confuse with mass effect

- Using Young water is in general confirmed by the isotopic study, but in this study using hydrodynamic and geochemical analysis

- I think it is good to minimise the text and synthesis the ideas in a conceptual model (geology, hydrogeology, geochemistry, hydrodynamic... of karst water) and to add a TDS spatial repartition of your karst water

- Added some recommendations

- Revise the english

Round 2

Reviewer 2 Report

In the table 1 (the values of Temperature with . or ,

Homogenize the reference list in the text X et al., or X et al.

some time in the text you must write (exp: Lorette et al. (2020) ).

also in the references list

Author Response

Dear Reviewer,

We have corrected these typographic errors in the Table 1 and the references.

We thank you for reviewing our work.

Best Regards,

David Viennet
